# Seasonal Effect of Grass Nutritional Value on Enteric Methane Emission in Islands Pasture Systems

**DOI:** 10.3390/ani13172766

**Published:** 2023-08-30

**Authors:** Helder P. B. Nunes, Cristiana S. A. M. Maduro Dias, Carlos M. Vouzela, Alfredo E. S. Borba

**Affiliations:** Institute of Agricultural and Environmental Research and Technology, Faculty of Agricultural and Environmental Sciences, University of the Azores, Rua Capitão João d’Ávila, 9700-042 Angra do Heroísmo, Açores, Portugal; cristianarodrigues@gmail.com (C.S.A.M.M.D.); carlos.fm.vouzela@uac.pt (C.M.V.); alfredo.es.borba@uac.pt (A.E.S.B.)

**Keywords:** methane conversion factor, ruminant nutrition, enteric methane, grass, emission factor

## Abstract

**Simple Summary:**

This study investigates the impact of seasonality on enteric methane emissions in islands with pasture systems, focusing on the 2006 Intergovernmental Panel on Climate Change (IPCC) Tier 2 methodology, refined in 2019. Feed samples for Azorean bovine were collected throughout the year, and their nutritional value and digestibility were determined. Significant differences were found between winter and summer pastures, with autumn presenting better nutritional quality. The total volume of enteric methane produced in the Azores was 20,341 t of methane (CH_4_), with peak emissions reaching 5837 t CH_4_ during the summer. Breeding bulls, beef cows, and heifers produce the highest amount of methane per animal, while pregnant dairy cows had the highest CH_4_ emissions per year, due to the high number of dairy cows in the archipelago. The study suggests that pastures are better managed during the autumn, resulting in lower emissions of enteric methane into the atmosphere.

**Abstract:**

Quantifying entericCH_4_ from grazing systems is a challenge for all regions of the world, especially when cattle feed mostly on pasture throughout the year, as pasture quality varies with the seasons. In this study, we examine the influence of seasonality on enteric methane emissions in the Azores, considering the most recent IPCC updates, to minimise errors in estimating enteric methane emissions in this region. For this purpose, samples of corn and grass silage, different types of concentrate, and pasture were collected throughout the year, and their nutritional value and digestibility were determined according to standard conventional methods. The estimation of methane production was conducted using the 2006 IPCC Tier 2 methodology, refined in 2019. The results revealed significant differences (*p* < 0.05) between the chemical composition of winter and summer pastures. However, it was in the autumn that these pastures presented the best nutritional quality. We estimated that the total volume of enteric methane produced in the Azores was 20,341 t CH_4_, with peak enteric methane emissions (5837 t CH_4_) reached during the summer. Breeding bulls, beef cows, and heifers are the categories that produce the highest amount of methane per animal. However, if we consider the total number of animals existing in the region, pregnant dairy cows are the category of cattle with the highest emissions of CH_4_. Thus, considering the current system of cattle production in the region, we can infer that the pastures are better managed during the autumn, which translates into lower emissions of enteric methane into the atmosphere during this season.

## 1. Introduction

Ruminants possess the ability to convert inedible food into high-quality food for human nutrition [1], such as milk and meat, ensuring food security worldwide [2]. However, animal agriculture is considered one of the main anthropogenic sources of greenhouse gas emissions [3]. Ruminants are the primary contributors to CH_4_ production in agriculture, mainly due to the emission of enteric CH_4_, which naturally arises from the process of rumen fermentation of feed [4]. In addition, the levels of CH_4_ emitted represent an energy loss for ruminants, which varies between 2 and 12% depending on various factors, including the type of feed ingested [5]. It is, therefore, crucial to accurately determine the emissions from ruminants in each region. This will enable the development of effective policies and informed decision making based on concrete data, ensuring compliance with standard international guidelines for national greenhouse gas inventories. Numerous research studies have been conducted to assess enteric methane production in various categories of cattle, with the majority being performed in confinement systems. Only 9% of the papers published in this field were performed under pasture conditions [2]. These studies indicate that in regions where cattle production is carried out through direct grazing, the emission rate of CH_4_ per unit of product is higher compared to mixed or confined production systems [6,7]. However, in grazing systems, determining the methane conversion factor (Ym; % of gross energy) precisely is challenging, which, in turn, can influence the calculation of the CH_4_ emission factor (EF; kg CH_4_/head/year). This uncertainty can result in the overestimation or underestimation of the actual enteric CH_4_ EF in cattle [8]. Consequently, when estimating the CH_4_ EF, many countries default to the value of Ym (6.5) in Tier 2 of the Intergovernmental Panel on Climate Change (IPCC) [9]. The Azores archipelago, located in the middle of the North Atlantic between latitudes 37° and 40° N and longitudes 25° and 31° W, is composed of nine islands, and has a territorial area of 223,196 ha. It presents excellent climatic conditions and fertile soils of volcanic origin, enabling direct grazing of cattle throughout the year [10]. Since 2016, the Regional Inventory of Emissions by Sources and Removals by Sinks of Air Pollutants (IRERPA) has been produced annually, where they estimate the production of enteric CH_4_ from cattle based on the Tier 2 methodology [11]. However, the digestibility of each type of feed consumed is assumed to be the standard digestibility as well as the methane conversion factor (% of EB converted to CH_4_) of the feeds, which was 6.5. In 2019, to determine ruminant emissions more accurately in various regions and meet standard international guidelines for national greenhouse gas inventories, the IPCC updated some of the standard equations and references. One of the updated parameters was Ym, noting that it is good practice for each country to determine its own Ym values, considering its herds and typical feeding characteristics. The IPCC also updated the Ym reference values, considering the general feed characteristics and production practices of various countries [12]. This update incorporated more detailed data on feed quality, particularly regarding Neutral Detergent Fibre (NDF) content and digestibility percentage.

In addition to updating the Ym reference values, the nutritional value of pastures differs with the seasons of the year, directly influencing the Ym value and, consequently, the EF. An assessment of the Ym is needed for each season of the year to produce more accurate methane emission estimates for each bovine category. To our knowledge, for this region, the influence of seasonality on the EF, according to different values of Ym in each season of the year, has not been studied yet. Therefore, this study represents the first investigation into the influence of seasonality on enteric methane emissions in the Azores. It considers the most recent IPCC updates concerning forage quality, Ym, and different bovine categories, aiming to minimise errors in the estimation of enteric CH_4_ emissions.

## 2. Materials and Methods

The present study was carried out in the Azores archipelago, where the production system is predominantly semi-extensive. In 2019, more than half of the total area of the archipelago (123,793 ha) was used as an agricultural area, with approximately 98% of this area devoted to cattle rearing and feeding. The agricultural area of the Azores is divided into four categories: arable land, without irrigation; grasslands; agricultural land, with natural vegetation areas; and natural grasslands, as presented in Figure 1. Permanent grasslands represent 43% of the region’s land area [13], mostly located at medium and high altitudes, with sub-spontaneous or even semi-natural grasslands, where grazing is less intensive [14]. Most of the arable land and improved pastures are in low-altitude and medium-altitude areas. Most of these pastures are sown with fodder maize in spring, to be harvested in late summer, which is preserved in the form of silage to be used as a grazing supplement during longer periods of grassland shortage. A total of 300 pasture samples representative of the Azorean grasslands were collected between autumn 2020 and summer 2021. The samples were collected according to the methodology presented by [14]. Briefly, 25 samples were taken from three different altitudes, low (below 200 m), medium (between 200 and 400 m), and high (above 400 m) each season (25 samples × 3 altitudes × 4 seasons), to ensure the variability and heterogeneity of the pastures ingested by the cattle. Samples were collected manually about 15 cm above the ground and transported to the Animal Nutrition Laboratory of the Agrarian Sciences Department of the University of the Azores, located in Angra do Heroísmo, Terceira, Azores, Portugal, where laboratory analyses were performed. The samples consisted of several plant species, with *Lolium perenne, Lolium multiflorum*, and *Trifolium repens* being the dominant species in the improved pastures. In the sub-spontaneous and/or semi-natural pastures, the predominant species were *Holcus lanatus, Lotus pedunculatus*, or *Poa trivialis*.

### 2.1. Characterisation of Bovine Farms

The region has approximately 293,000 bovines [15] and an annual milk production rate of about 652 million litres [13]. Livestock production farms in the Azores are characterised by small farms, typically covering an area of less than 50 ha, and are divided into small, discontinuous paddocks, each usually ranging between 0.1 and 0.5 ha [16]. Cattle move between the farms’ pasture paddocks, grazing them directly throughout the year. Nevertheless, dairy cows consume grass for 365 days per year; they are usually supplemented with corn silage and grass, and concentrate is added to the diet when milking takes place. Beef cattle usually feed on pasture, and, in some periods of lower pasture production, their feed is supplemented with grass silage.

In this work, the cattle were primarily sorted into groups according to their intended function and age. When more detailed data were available, such as gender or pregnancy status for females, these were included, thus creating 10 categories: beef calves, dairy calves (female and male), replacement heifers, beef cattle (pregnant or not), dairy cattle (pregnant or not), breeding bulls, and other cattle (Figure 2). Every detail is important to determine the energy consumption in each category. The “Other bovines” category included all animals intended for slaughter and over one year old, regardless of fitness or breed. Generally, these animals are confined to smaller parcels of land and are fed a higher energy content. While not common in the Azores’ production system, there are some very sporadic cases where the animals are kept in feedlot systems. These have also been included in this category.

In the dairy sector, the Holstein Friesian breed predominates in the Azores. However, other dairy breeds have recently been introduced, such as the Jersey breed, which have better grazing adaptability and high-fat content in milk. In the meat sector, there are pure meat breed centres, such as Limousine, Aberdeen Angus, Charolais, and Simmental Fleckvieh. Nevertheless, most animals with meat aptitude come from crosses between these breeds and animals of dairy aptitude. Since it was not possible to obtain data on the number of animals the different categories, we opted to use the average values published by the official entities of the region.

### 2.2. Determination of Nutritional Parameters

#### Chemical Analyses

After being collected, the samples were dried at 65 °C in an oven with forced air circulation until a constant weight was reached [14,17]. They were then cut into small pieces and ground with a Retsch mill (GmbH, Hann, Germany), sieved using a 1 mm sieve, and stored in tightly closed bags. For chemical determination of the samples, the Weende system was used to determine the Dry Matter (DM, method 930.15), Crude Protein (CP, method 954.01), Ether Extract (EE, method 920.39), and Total Ash (method 942.05) according to the standard methods of [18]. The Neutral Detergent Fibre (NDF), Acid Detergent Fibre (ADF), and Acid Detergent Lignin (ADL) were determined according to the method used by [19].

### 2.3. Determination of Biological Parameters

The biological parameters, more specifically, in vitro dry matter digestibility and organic matter digestibility, were determined using the method described by [20], with modifications outlined in [21]. The rumen liquid used in the determinations was collected from a local bovine slaughterhouse, following the procedure described in [14,22]. The conditions for obtaining rumen liquid were detailed in [22], with rumen samples collected from five healthy dairy cattle (Holstein-Friesian) that had been fed ryegrass. Once collected, the rumen fluid was preserved at 38 °C under anaerobic conditions and delivered to the animal nutrition laboratory within 30 min.

### 2.4. Development of Tier 2 Enteric Methane Emission Factors for Cattle in the Azores

We used Tier 2 methodology developed at the IPCC 2019, a version improved from the IPCC 2006 [11], to calculate the enteric fermentation of CH4 released by cattle in the Azores archipelago [11]. This methodology was adopted because there are specific data for cattle production in the Autonomous Region of the Azores (RAA), such as milk production (dairy and beef cows), milk fat content, growth rate (calves), time spent in “stabling/grazing”, the proportion of pregnant cows per year (dairy and beef cows), and the nutritional value and digestibility of the consumed feed.

To estimate the total emission of enteric methane (EM_CH_4__) produced by cattle in the Azores, the following equation was used:(1)$${EM}_{{CH}_{4}total}=\frac{{No\;of\;animals}_{t}\times {\Sigma EF}_{t}}{1000}$$
where, ${EM}_{CH4total}$ are the methane emissions from enteric fermentation (t CH_4_/year); No of animals corresponds to the number of animals in each category t; ${EF}_{t}$ is the enteric fermentation methane emission factor of category t (kg/head/year).

For a more accurate estimate of methane, the animals were grouped into 10 different categories (t), with feed adjusted to each category and according to the season of the year. Thus, the enteric fermentation methane emission factor (EF) was estimated for each category, based on the seasonal nutritional value of the pasture, gross energy intake (GEI) and CH_4_ conversion rate Ym (%) as follows:(2)$${EF}_{t}=\frac{GEI\times \left(\frac{Ym}{100}\right)\times days/year}{55.65}$$
where ${EF}_{t}$ is the CH_4_ emission factor (kg CH_4_/head/year). The GEI is the gross energy intake (MJ/d). The Ym represents the CH_4_ conversion rate (%), which denotes the fraction of gross energy in feed converted to CH_4_ (CH_4_ yield). Ym is variable depending on feed quality and digestibility. In this study, Ym was estimated according to the tables published by [12]. The days/year parcel is the number of days per year that the animal is exposed to a type of feed. The factor, 55.65 (MJ/kg CH_4_), is the energy content of the methane.

In regions like the Azores, the predominant system used throughout the year is direct grazing of pastures, making it challenging to determine the gross energy intake (GEI). Therefore, to estimate GEI, considering the specificity of the RAA, to estimate the following equation was employed:(3)$$GEI=\frac{\left(\frac{NEm+NEa+NEl+NEp}{REM}\right)+\left(\frac{NEg}{REG}\right)}{\frac{DE}{100}}$$
where, GEI is the gross energy intake (MJ/head/day); NEm the net energy for maintenance (MJ/day); NEa the net energy for activity (MJ/day); NEl the net energy for lactation (MJ/day); NEp the net energy for pregnancy (MJ/day); NEg the net energy for growth (MJ/day); DE% is the digestible energy expressed as a percentage of gross energy; REM the ratio of net energy available in a diet for maintenance to digestible energy consumed; and REG the ratio of net energy available for growth in a diet to digestible energy consumed.

According to [11], one should also consider the net energy expended by the animal on agricultural or traction work. However, in this study, this parameter was not estimated, as the energy expenditure by cattle in this category in the Azores is currently considered negligible.

The NEm was calculated as:(4)$$NEm=Cfi\times {\left(Weight\right)}^{0.75}$$
where, NEm represents the net energy for maintenance (MJ/day); Cfi is maintenance coefficient; and Weight is the live animal weight in kg.

The NEa is the net energy expended by animals in obtaining food, water, or shelter. It depends more on how the animal feeds than on the food itself, and is estimated using the equation:(5)NEa=Ca×NEm
where, NEa is the net energy spent on activity (MJ/head/day); Ca is the coefficient corresponding to the feeding situation of the animal, and NEm is the net energy for maintenance (MJ/day).

The net energy for lactation (NEl) is expressed in MJ/day. NEl is the net energy required for animals to produce milk during the lactation period. For dairy cows in this study, the lactation period was 305 days, whereas for beef cows, it was 190 days.
(6)NEl=Pm×(1.47+0.40×Fat)
where, NEl is net energy for lactation (MJ/head/day); Pm is the daily milk production (kg/head/day) and Fat is the milk fat content percentage.

The NEp is the extra net energy needed during the pregnancy phase of cows. It was estimated using the equation:(7)NEp=Cp×NEm
where NEp is the net energy for pregnancy (MJ/day), Cp represents the pregnancy coefficient, and NEm is the net energy for maintenance (MJ/day).

NEg represents the net energy spent by the animal in growth, that is, in weight gain. This variable was only calculated for the “calves” subcategory. It was estimated using the equation:(8)$$NEg=22.02\times {\left(\frac{BW}{C\times MBW}\right)}^{0.75}\times {WG}^{1.097}$$
where NEg is the net energy for growth (MJ/day); BW designates the body weight (kg), C is the growth coefficient, MBW is the mature body weight (kg), and WG the average daily weight gain (kg/day).

The REM was calculated as:(9)$$REM=\left[1.123-\left(5.16\times 10^{-3}\times DE\right)+(1.26\times 10^{-5}\times {\left(DE\right)}^{2}-\left(\frac{25.4}{DE}\right)\right]$$
where REM is the ratio of net energy available in a diet for maintenance to digestible energy consumed; and DE is the digestible energy, expressed as a percentage of gross energy.

The REG was calculated as follows:(10)$$REG=\left[1.164-\left(5.16\times 10^{-3}\times DE\right)+(1.308\times 10^{-5}\times {\left(DE\right)}^{2}-\left(\frac{37.4}{DE}\right)\right]$$
where REG is the ratio of net energy available for growth in a diet to digestible energy consumed; DE represents the digestible energy expressed as a percentage of gross energy.

### 2.5. Statistical Analyses

The data were analyzed statistically using SPSS Statistics Software v.27 (IBM SPSS, Inc., Chicago, IL, USA). The statistical significance of the difference between the distributions was evaluated for normality using the Shapiro–Wilk test, and the homogeneity of variance was assessed using Levene’s test. For the comparison of multiple independent groups with normally distributed data, we employed one-way ANOVA, followed by post hoc testing using Duncan’s multiple range test to determine significant differences. Comparisons were considered statistically significant when the *p*-value was lower than 0.05.

## 3. Results

### 3.1. Determination of Nutritional Parameters

The factors influencing pasture growth vary throughout the year, leading to fluctuations in its nutritional value across seasons. In Table 1, we can see the variation of the different nutritional parameters during each season. In autumn, the pasture exhibits a lower DM content (9.42%) and a higher protein content (22.91% DM), showing significant differences from summer and spring. On average, the NDF value is 68.19% DM and the ADF is 33.22% DM, with the highest NDF and ADF values observed in the summer, reaching 76.71% DM and 37.22% DM, respectively.

Regarding the biological parameters, namely, the in vitro digestibility of dry and organic matter (Table 1), we observed that the pasture exhibited higher digestibility in autumn, while in summer, it showed the greatest digestibility. Notably, in terms of organic matter digestibility, significant differences (*p* < 0.05) were observed only during summer.

Although dairy and beef cattle primarily feed on pasture, they also receive supplementation with concentrate, which varies according to the aptitude and category of the animal, along with corn silage and pasture. The average nutritional values of each component comprising the diet are presented in Table 2.

### 3.2. Diet Composition

For each category of bovine, a base diet was estimated which reflects the percentage of the diet’s composition (Figure 3) and includes pasture, grass silage, corn silage, and concentrate.

We can see that, regardless of the season, pasture, whether fresh or preserved as silage, is the basis of food for Azorean cattle in all categories. It should be noted that fresh pasture is present in all categories, even in the “Other cattle” category, which has a lower consumption of pasture, since this category includes animals that are being fattened for slaughter. The “dairy cows” category includes both pregnant and non-pregnant cows and is one that exhibits the most significant variation in feeding. During the summer, particularly in dairy cows, there is a reduction in pasture consumption due to its scarcity and lower quality. Consequently, cows are supplemented with grass silage and concentrate. In autumn and winter, additional corn silage and concentrate are included in the dairy cattle feed to provide them with more energy.

### 3.3. Enteric Methane Emission Factors for Cattle in the Azores

#### Coefficients Used

To estimate the methane emission factor of cattle in the Azores archipelago, it is necessary to resort to previously estimated coefficients. The maintenance (Cfi), activity (Ca), growth (Cg), and gestation (Cp) coefficients were estimated according to the 2019 refined 2006 IPCC Tier 2 methodology [12,23]. The coefficient values, shown in Table 3, were found to be the most appropriate for each bovine category, according to the Azorean livestock production system.

Regarding the growth coefficient, we note that the NRC (2001) suggests a value of 0.8 for females and 1.2 for males. As the official data on which this study were based did not provide data on animals by gender in the categories, “Beef Calves” and “Other Bovines”, we assumed an equal distribution of 50% females and 50% males in both categories. To determine the growth coefficient for these two categories, we calculated the average with the coefficients for males and females, and the value found was 1.

### 3.4. Estimation of Enteric Methane Emission Factors

The values of NEm, (MJ/day), NEa (MJ/day), NEg (MJ/day), NEl (MJ/day), NEp (MJ/day), DE (as % of GE) REM (%), REG (%), GEI (MJ/kg), and Ym (%), were calculated for each cattle category and are presented in Table 4. To account for the unique nutritional value of the pasture in each season and the diet of each category of cattle, we calculated DE, REM, REG, and GEI specifically for each season.

Based on the digestibility and NDF value of each meal during each season, the value of Ym was calculated for spring, summer, autumn, and winter, adhering to the recommendations Table 10.12 of the 2019 refined 2006 IPCC Tier 2 methodology [11,12].

During summer, bovines exhibit a higher emission of enteric CH_4_, with an average of 20.01 kg CH_4_, whereas the minimum emission occurs in autumn at 15.55 kg CH_4_. In Table 5, we can see in more detail the amount of CH_4_ emitted per animal in each category. The “Breeding Bulls” category emits, per animal, 79.09 kg CH_4_, followed by the beef cows with 76.64 kg CH_4_. On the other hand, the cattle category that emits, per head, the least amount of CH_4_ throughout the year (45.39 kg CH_4_), is “Other Bovines”.

In absolute terms, the lowest emission per head is reached in autumn (9.82 kg CH_4_) in the “Other bovines” category, a value very similar to the one found in winter, with 9.83 kg CH_4_/head recorded in the same bovine category. Conversely, the absolute maximum of CH_4_ emissions was observed during the summer in the “Breeding bulls” category, with each animal emitting 23.71 kg of CH_4_.

The estimated total CH_4_ emissions by category and season for the Azores are presented in Table 6. Annually, it is estimated that 20,341 t of CH_4_ are emitted from the enteric fermentation of cattle, with dairy cows being the category responsible for the largest amount of enteric CH_4_ emissions.

Overall, global seasonal methane production exhibits a variation of 1200 t CH_4_, with the highest emissions occurring during the summer and the lowest being recorded in the autumn. Examining each category individually, we find that the “Dairy cattle—Pregnant” category emits 5270 t CH_4_ per year, with peak production occurring in summer (1446 t CH_4_) and the lowest emission value in autumn (1245 t CH_4_). The category that emits the least CH_4_ is the “Breeding bulls”, which emit 86 t CH_4_ in autumn and 87 t CH_4_ in winter; annually, the emission of this category is 395 t CH_4_.

## 4. Discussion

Estimating enteric CH_4_ emissions in grassland systems remains a significant challenge for all stakeholders due to the considerable variability of results based on real data. Despite efforts to make enteric methane measurement techniques better [17], results still lack consistency. This is due to variables including the unit, production, and animal category under consideration. To the best of our knowledge, this is the first attempt to estimate enteric CH_4_ production for each season in the Azores, considering the different cattle categories as well as the determination of the nutritional value of the diet and its digestibility for each category. The information obtained from this study is crucial for understanding the levels of enteric CH_4_ emissions in the region and for implementing sustainable measures in livestock production. Given that this economic activity is one of the pillars of the Azorean economy, it is essential to address its impact on the environment, as it represents the main source of CH_4_ and N_2_O emissions in the region [24].

The chemical composition, especially the dry matter and fibre content, as well as the digestibility present in feed, is directly related to the production of enteric CH_4_ by ruminants [25]. Some researchers suggest increasing the intake of more digestible forages as the main measure to mitigate CH_4_ emissions [7]. However, [26] reports that increasing forage digestibility results in an increase in dry matter intake and consequently, an increase in CH_4_ emissions by cattle. In recent years, several methodologies and equations have been developed to improve the estimation of CH_4_ emission in pasture-based systems. However, it is crucial to incorporate the nutritional value of pastures and forages into these equations and methodologies since their chemical and biological compositions vary throughout the year [14], thus influencing CH_4_ production.

Bovine production in the Azores is based on pasture, with cattle rotating between different plots throughout the year in a kind of transhumance. However, pasture production is not consistent year-round, with an excess of grass production in spring and a reduction during two seasons: summer, particularly in the lowlands of the islands, and winter, especially in areas at higher altitude [21,27].

Due to the seasonality of grass production, the natural nutritional variability throughout the year, the pasture and forage management, and the cattle production system in the region (semi-extensive regime), the estimation of enteric CH_4_ emissions becomes even more complex. In our study, we accounted for all the variables required for estimating enteric CH_4_ in the Azores archipelago, following the methodology from IPCC Level 2 of the 2019 Refinement to IPCC 2006. One of the crucial variables for calculating the amount of enteric CH_4_ emission factor (EF) per animal (kg CH_4_/head) and the percentage of gross energy intake used for CH_4_ conversion is represented as Ym. However, the value of Ym is poorly documented for combinations of different feeds and pastures in different seasons [9], necessitating the estimation of Ym for each country or region to mitigate potential errors in EF CH_4_ estimates [28]. As referenced by [12], Ym for bovines is associated with the quantity and quality of feed, specifically the NDF content and the percentage of diet digestibility. For dairy cows, in addition to the factors mentioned above, the IPCC, 2006 guidelines note that Ym is also influenced by annual milk production levels, with a reference value of 6.5% for low-producing dairy cows. After the 2019 revision, the IPCC linked dietary NDF content and feed digestibility, with value of 6.5% only being applied when animal diets have a feed digestibility of less than 62% and NDF fractions greater than 38%. However, for non-dairy cattle, the value of Ym is dependent only on the percentage of digestibility. In diets where animals are not confined, distinctions can be made between forage-based diets, for which the Ym value of 7.0 should be used, and for mixed-concentrate diets or high-quality forage diets, for which the IPCC 2019 Tier 2 recommends a Ym value of 6.3%, i.e., if the diet has a DE% value between 62–71% [12].

In this study, for the calf category, although the IPCC 2019 recommends that enteric CH_4_ emissions should be zero while they are only fed milk, we assumed a Ym value of 6.3%, since the ages in the calf category range from 0–12 months. However, despite all the recommendations, the IPCC 2019 warns that when Ym is not calculated, a thorough knowledge of the production and feeding systems of each region should be considered when making the decision to choose the most appropriate Ym value.

Our data on the chemical and biological composition of pastures (Table 1) revealed significant variability between the four seasons, with the most significant differences observed between summer and the other seasons in all analysed parameters. This variability in pastures is reflected in the nutritional value of the final diet, since most cattle categories rely on pasture as their primary food source, as shown in Figure 3.

A detailed analysis of the chemical and biological composition of the pastures in summer revealed that the DM content was 24.89% (*p* < 0.01) and the NDF value was 76.7% DM (*p* = 0.02), both significantly higher compared to the other seasons, and the low protein content (11.63% DMP = 0.03) and low dry matter digestibility (46.71%; *p* < 0.05) differed statistically from the other seasons. During summer, due to lower availability of pastures in low- and medium-altitude areas, farmers move bovines to graze in medium-high and high-altitude areas, i.e., pastures situated at 400 m above sea level. These pastures have a more rustic floristic composition, and in some cases, they are composed of predominantly natural or sub-spontaneous species of lower nutritional quality. With longer days and higher temperatures, pastures mature faster, gaining a higher fibre content [29]. The combination of these two factors make the pastures richer in fibre during the summer, with a lower cellular content and decreasing digestibility, as documented by the results obtained in this work. To remedy this situation, there is an increase in the supplementation of cattle with conserved forages (grass and corn) in the form of silages and animal concentrates (Table 2). The low quality of pastures in summer significantly impacts the overall diet quality, leading us to indicate a Ym value of 6.5% for the category of dairy cows (pregnant and non-pregnant), using [12] as a reference, but it is not enough to improve digestibility levels and thus improve Ym. For the remaining categories, where there is a high dependence on pasture, the established Ym value was 7%.

After the summer, the animals transition to grazing in coastal areas (below 250 m altitude) where temporary pastures predominate. These pastures primarily consist of varieties of *Lolium multiflorum*, offering improved nutritional value and higher digestibility, thereby influencing the findings of this study. Permanent pastures in the middle zones (between 250 and 400 m) of the archipelago are predominantly composed of *Lolium perenne* and a variety of species from the *Trifolium* genus. During the spring, thanks to the improved weather conditions and increased photoperiod, there is an abundance of grass production in the region. The surplus is mainly collected and preserved as silage, to be used during periods of limited pasture availability. It is also during spring that part of the low- and medium-altitude land is prepared for the sowing of maize, which will be harvested at the end of the summer, and which will serve as an energy source for cattle feeding throughout the year. The obtained results after the determination of the nutritional value of maize and grass silages (Table 2) are in line with those reported by [14] for the Azores archipelago for this type of food. In general, pastures greatly improved their nutritional quality between autumn and spring. During this period, the lowest DM levels were observed in the autumn, gradually increasing until late spring. The NDF content of pastures reached its minimum in winter (63.92% DM), but there were no significant differences between autumn, winter, and spring. ADF and ADL contents were significantly higher in spring (*p* < 0.05) compared to autumn, due to the rapid growth and maturation of plants in this season [30]. Regarding digestibility, the highest value found in pastures was 65.16% DM during the autumn. However, in spring, with the increase in fibre content in pastures, the digestibility is lower (54.98% DM), with this difference between the mean values being statistically significant (*p* < 0.05). Due to this change in the chemical and biological composition of the pasture, the digestibility value and NDF content of the diet also changed, leading to the derivation of new Ym values. In dairy cows, between autumn and spring, diets have digestible fractions ranging between 63% and 70% and an NDF content of more than 37%, with associated milk production of between 5000 and 8500 kg/year. Thus, according to the recommendation of [12], the value of 6.3% for Ym was used for these seasons. The estimation of EF depends essentially on the nutritional quality of the cattle diet, each category’s animal characteristics (Appendix A), the production system, the methane conversion rate, and the gross energy intake of each category of cattle in the various seasons.

Due to seasonal variation in cattle diet, primarily influenced by the nutritional value of pastures, pasture management, and availability, enteric methane production is not consistent throughout the year across all categories, with the highest emissions per head occurring during the summer. The results showed that the categories that emit the most CH_4_ per head to the atmosphere are Breeding Bulls (79.09 kg CH_4_/head/year), Replacement Heifers (75.19 kg CH_4_/head/year), and Beef Cattle-Pregnant (76.64 kg CH_4_/head/year), with these categories being the ones that consume the largest amount of pasture. Conversely, “Other Bovines” was the most efficient category, emitting a lower amount of CH_4_ of enteric origin. Higher-quality feed that is less susceptible to variations during the year is more efficient, resulting in lower amounts of CH_4_ being produced [31]. Our findings revealed that in the Azores, the average methane emissions per head per year for each bovine are about 68.8 kg CH_4_. In 2019, IRERPA reported an average emission of approximately 77 kg CH_4_ per bovine throughout the year. In New Zealand, where grazing conditions resemble those in the Azores, the average emission for each cattle is about 79.5 kg of CH_4_ per year [28]. In a study conducted by [32], which addresses the environmental and economic impact of greenhouse gas production in Azorean dairy production, estimates showed that each dairy cow emits 115.5 kg CH_4_ per head per year, a relatively high value when compared to the result obtained in our present study (71.34 kg CH_4_ per head per year) and more recently indicated by [24] at 94 kg CH_4_ per head per year. The total CH4 emission values were different for each season. Summer stood out as the season with the highest CH_4_ emissions, with 5837 t CH_4_, mainly due to the low quality of the pasture present in the cattle diet. However, this value could have been even higher if the animals were fed exclusively on pasture. What we observe is that during this season, most producers compensate for the lower quality of pastures by supplementing the animals with higher-quality feed, such as concentrates and/or grass silage.

In autumn, we estimate that around 4637 t CH_4_ are emitted, marking the lowest value throughout the year. The reduction in CH_4_ emissions during autumn compared to summer is primarily attributed to cattle feed. During this season, they graze on improved pastures, which are richer in nutrients and offer better digestibility. In addition, with the decrease in photoperiod and lower temperatures, grass growth is slower, allowing producers to better manage pastures.

When it comes to winter, although pastures have the lowest digestibility throughout the year, climatic conditions are not as favourable for grass production, essentially due to the high amount of precipitation and persistent humidity levels above 90%. This leads to scarce pasture availability, making it necessary to resort to supplementation with grass silage and, especially, maize silage, produced at the end of summer, leading to a change in the ruminants’ diet during this season. However, despite these dietary changes, the levels of CH_4_ emitted (4674 t CH_4_) are comparable to those observed in autumn. In the case of dairy cows, which comprise over 30% of the cattle population, it is common to increase the amount of concentrate fed during winter, exceeding 8 kg per animal, to fulfil their energy needs. The reinforcement of concentrate in the diet is a response to the existence of numerous calving’s that occur during the winter, which is common practice in the Azores. This timing aims to align the peak of lactation with the period of higher pasture production in early spring [33]. The inclusion of high levels of concentrate in the diet of dairy cows, exceeding 8 kg per day, leads to a reduction in Ym and, consequently, the amount of CH_4_ emitted [34]. Furthermore, the type and quantity of concentrate supplementation in different categories and grazing management are also factors that directly influence the seasonal estimation of EF CH_4_ [35]. In early spring, pasture reaches its peak production, leading to reduced reliance on concentrate and silage supplementation in dairy cow diets. Similarly, in the remaining categories, there is an increase in the proportion of pasture in the cattle diet compared to silage supplementation. However, as the season progresses, the nutritional value of the pasture starts to decline and it is necessary to adjust the feeding. During this time, the amount of maize silage available is lower, and it is mainly reserved for dairy cows. The combination of these two factors results in a diet richer in NDF, with a lower protein content and lower digestibility when compared to autumn and winter. This promotes greater fermentation by methanogenic bacteria and, consequently, greater production of enteric CH_4_ [36]. Thus, during the spring, 5194 t CH_4_ were emitted, with the main contributions coming from the “dairy cows (pregnant and non-pregnant)”, “replacement heifers”, and “beef cows (pregnant and non-pregnant)” categories. These three categories are the ones with the highest number of animals, as can be seen in Figure 2, with dairy cattle accounting for most of the bovine population (32%), followed by beef cattle and heifers, both at 17%. Consequently, they are also the categories responsible for the highest overall CH_4_ emissions. It should be noted that, in the Azores, most producers place replacement heifers to graze on marginal land or in pastures with lower nutritional value throughout the year. As a result, these heifers are typically supplemented with grass silage during periods of reduced pasture availability.

In total, our estimation indicates that cattle in the Azores emit approximately 20,341 t CH_4_ per year, which is lower than the estimate of 21,462 t CH_4_ reported by [24]. This variation in results highlights the significance of understanding the seasonal nutritional value of pastures and their digestibility in each location. Such knowledge enables us to more accurately determine the amount of CH_4_ emitted by each region.

## 5. Conclusions

Our estimates of enteric CH_4_ emissions for cattle in the Azorean pasture system, based on the IPCC Tier 2 equations from the 2019 refinement to IPCC 2006, reveal that summer is the season in which cattle emit the highest amount of enteric CH_4_. This can be attributed primarily to the lower quality of pasture during this time of the year. On the other hand, in autumn, due to appropriate supplementation and, especially, to improved pasture management, CH_4_ emissions reach their minimum. In production systems where animals feed directly on pasture all year round, it is crucial to control their nutritive value and digestibility to accurately estimate parameters such as Ym, ED and, GEI, which play a fundamental role in estimating the EF. These findings challenge policymakers and cattle producers to reconsider their choices regarding the type of pasture and forage production, their management practices, and the overall cattle diet, with the aim of minimising enteric CH_4_ emissions without compromising animal welfare and productivity. Further research is warranted in this area, with a focus on obtaining more detailed data on cattle production, which would allow for adjustments and refinements in the estimates of enteric CH_4_ emissions while consolidating the results obtained so far.

## Figures and Tables

**Figure 1 animals-13-02766-f001:**
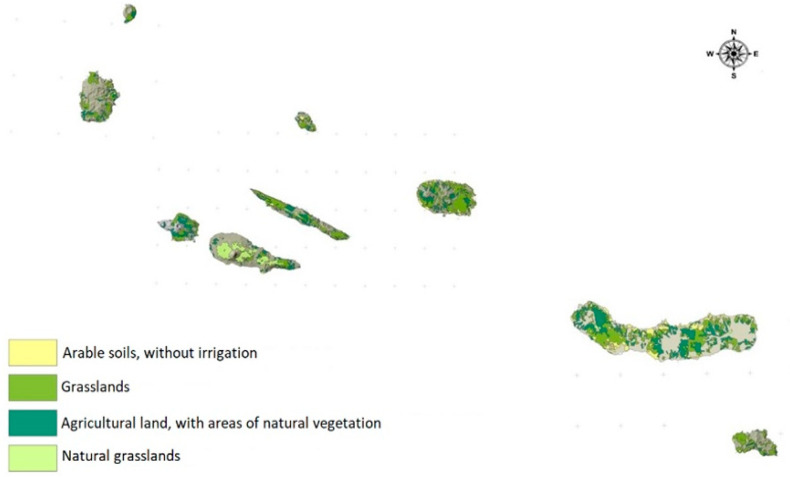
Geographical distribution of the agricultural area in the Azores.

**Figure 2 animals-13-02766-f002:**
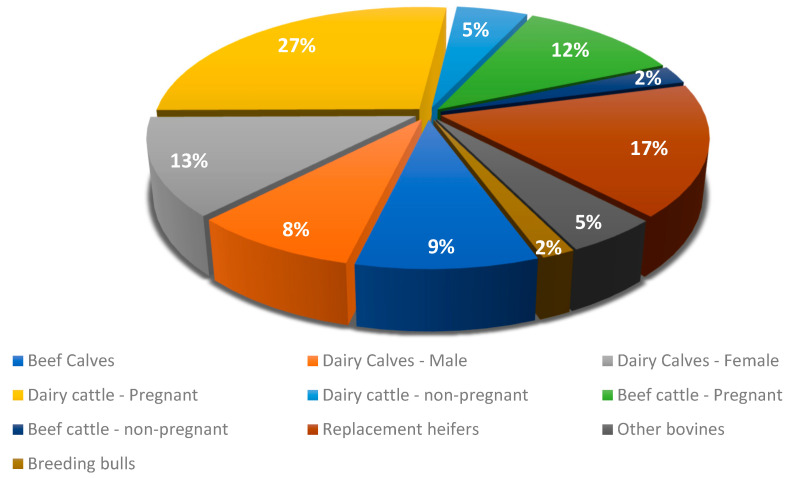
Distribution of bovines by category.

**Figure 3 animals-13-02766-f003:**
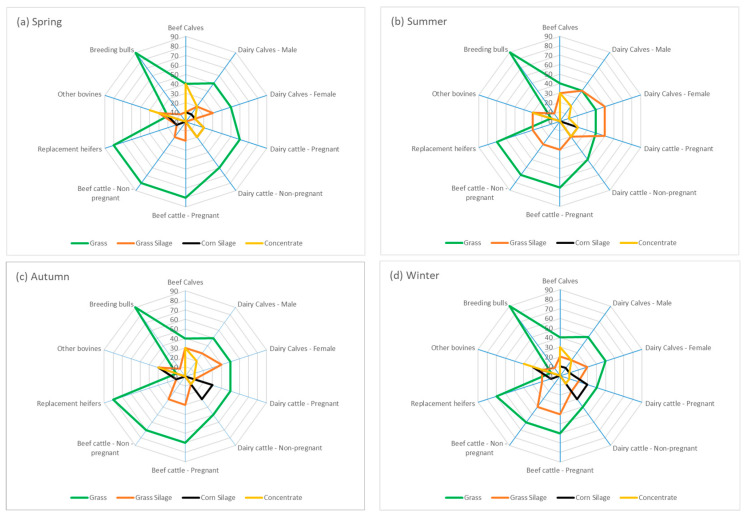
Diet composition for category and season.

**Table 1 animals-13-02766-t001:** Nutritional value of pasture throughout the year.

	Winter	Spring	Summer	Autumn	Mean	SEM	*p* Value
DM (%)	12.44 ^a,b^	16.89 ^b^	24.89 ^c^	9.42 ^a^	15.91	2.91	<0.01
CP (%DM)	21.58 ^a^	14.62 ^b^	11.63 ^b^	22.91 ^a^	17.69	2.35	0.03
NDF (%DM)	63.92 ^a^	66.22 ^a^	76.71 ^b^	65.90 ^a^	68.19	2.49	0.02
ADF (%DM)	28.01 ^a^	33.42 ^b^	37.22 ^c^	34.23 ^b^	33.22	1.66	<0.01
ADL (%DM)	2.25 ^a^	4.64 ^b^	4.95 ^b^	5.52 ^b^	4.34	0.62	<0.05
EE (%DM)	3.20 ^a^	2.20 ^b^	1.73 ^b^	1.81 ^b^	2.24	0.30	0.04
Ash (%DM)	12.94 ^a^	7.90 ^b^	7.36 ^b^	12.75 ^a^	10.24	1.31	<0.01
DMD (%)	64.02 ^a^	54.98 ^b^	46.71 ^c^	65.16 ^a^	57.72	3.74	<0.05
OMD (%)	57.73 ^a^	51.83 ^a^	44.30 ^b^	61.16 ^a^	53.76	3.20	<0.05

DM, dry matter; CP, crude protein; NDF, neutral detergent fibre; ADF, acid detergent fibre; ADL, acid detergent lignin; EE, ether extract, DMD, dry matter digestibility; %, percentage; %DM, percentage dry matter; Means with different letters in the same line are significantly different; SEM, standard error of mean.

**Table 2 animals-13-02766-t002:** Nutritional value of different concentrates and corn and grass silages.

Feeds	DM (%)	Per 100 g of DM						DMD (%)
		CP	NDF	ADF	ADL	EE	Ash	
Grass Silage	31.97	12.72	60.12	39.73	5.64	3.10	11.07	61.78
Corn Silage	31.24	7.64	49.40	31.50	5.41	3.20	5.98	70.02
Calves concentrate	86.89	18.70	25.27	7.22	2.03	3.91	6.79	81.98
Finishing concentrate	87.05	18.81	27.99	8.30	2.77	3.18	6.34	81.56
Heifers concentrate	87.85	18.93	26.78	11.69	2.35	3.57	6.38	80.84
Dairy cattle concentrate	91.24	18.78	26.97	14.19	2.24	3.21	6.57	85.01

DM, dry matter; CP, crude protein; NDF, neutral detergent fibre; ADF, acid detergent fibre; ADL, acid detergent lignin; EE, ether extract, DMD, dry matter digestibility; %, percentage.

**Table 3 animals-13-02766-t003:** Coefficients used to estimate CH_4_ emission factors from enteric fermentation in bovine categories using the 2019 refined 2006 IPCC Tier 2 methodology and source NRC, 2001.

Category	Coefficients			
	Maintenance (Cfi)	Activity (Ca)	Growth * (Cg)	Pregnancy (Cp)
Beef Calves	0.322	0.17	1	n.a
Dairy Calves—Male	0.322	0.17	1.2	n.a
Dairy Calves—Female	0.322	0.17	0.8	n.a
Dairy cattle—Pregnant	0.386	0.17	n.a	0.1
Dairy cattle—non-pregnant	0.386	0.17	n.a	n.a
Beef cattle—Pregnant	0.386	0.17	n.a	0.1
Beef cattle—non-pregnant	0.386	0.17	n.a	n.a
Replacement heifers	0.322	0.17	0.8	n.a
Other bovines	0.322	Na	1	n.a
Breeding bulls	0.37	0.17	n.a	n.a

n.a—not applicable; * Source: NRC, 2001. Cfi, coefficient of maintenance; Ca, coefficient of activity; Cg, coefficient of growth; Cp, coefficient of pregnancy.

**Table 4 animals-13-02766-t004:** Estimated net energy requirements, digestible energy, gross energy intake, ratios of net energy, and CH_4_ conversion rate by bovine category.

Parameter		Calves			Dairy Cattle		Beef Cattle		Replacement Heifers	Other Bovines	Breeding Bulls
		Beef Calves	Dairy Calves		Pregnant	Non-Pregnant	Pregnant	Non-Pregnant			
			Male	Female							
NEm (MJ/day)		17.12	15.49	15.49	43.84	37.71	45.62	43.24	28.80	36.57	50.35
NEa (MJ/day)		2.91	2.63	2.63	7.45	6.41	7.76	7.35	4.90	0.00	8.56
NEg (MJ/day)		8.07	4.65	6.30	n.a	n.a	n.a	n.a	11.71	26.90	n.a
NEl (MJ/day)		n.a	n.a	n.a	69.84	52.38	17.34	13.46	n.a	n.a	n.a
NEp (MJ/day)		n.a	n.a	n.a	6.98	5.24	1.73	1.35	n.a	n.a	n.a
DE (as %GE)	Spring	65.12	62.42	60.40	63.17	63.17	56.34	56.34	55.66	67.38	55.66
	Summer	63.32	59.79	56.27	61.22	61.22	51.23	51.23	49.72	67.91	49.72
	Autumn	69.19	67.51	65.49	68.27	68.27	64.15	64.15	64.82	70.92	64.82
	Winter	69.56	67.75	64.92	69.194	69.19	63.35	63.35	63.57	70.47	63.80
REG (%)	Spring	1.35	1.39	1.42	1.37	1.37	1.50	1.50	1.51	1.31	1.51
	Summer	1.38	1.43	1.50	1.41	1.41	1.60	1.60	1.63	1.30	1.63
	Autumn	1.28	1.31	1.34	1.30	1.30	1.36	1.36	1.35	1.26	1.35
	Winter	1.22	1.27	1.31	1.25	1.25	1.38	1.38	1.37	1.27	1.37
REM (%)	Spring	1.20	1.24	1.26	1.23	1.23	1.31	1.31	1.32	1.18	1.32
	Summer	1.22	1.27	1.31	1.25	1.25	1.38	1.38	1.41	1.17	1.41
	Autumn	1.16	1.18	1.20	1.17	1.17	1.21	1.21	1.21	1.14	1.21
	Winter	1.15	1.17	1.21	1.16	1.16	1.22	1.22	1.22	1.14	1.22
GEI (MJ/day)	Spring	39.97	34.39	37.33	200.77	152.77	127.86	113.67	74.48	84.70	105.84
	Summer	40.91	35.73	39.69	207.16	157.63	140.62	125.01	82.23	84.22	118.48
	Autumn	38.04	32.10	34.85	185.79	141.37	112.31	99.4	65.34	1.67	90.88
	Winter	37.87	31.99	35.11	183.29	139.47	113.72	101.10	66.43	82.04	92.35
Ym (%)	Spring	6.30	6.30	6.30	6.30	6.30	7.00	7.00	7.00	6.30	7.00
	Summer	6.30	7.00	7.00	6.50	6.50	7.00	7.00	7.00	6.30	7.00
	Autumn	6.30	6.30	6.30	6.30	6.30	6.30	6.30	6.30	4.00	6.50
	Winter	6.30	6.30	6.30	6.30	6.30	6.30	6.30	6.30	4.00	6.50

NEm, net energy for maintenance; NEa, net energy for activity; NEg, net energy for growth; NEl, net energy for lactation; NEp, net energy for pregnancy; DE, digestible energy; REG, ratio of net energy available for growth in a diet to digestible energy consumed; REM, ratio of net energy available in a diet for maintenance to digestible energy consumed; GEI, gross energy intake; Ym, methane conversion rate.

**Table 5 animals-13-02766-t005:** Estimation of emission factor by season and category.

	Emission Factor (Kg CH_4_/Head/Season)				Total per Year (kg CH_4_/Head)
	Spring	Summer	Autumn	Winter	
Beef Calves	16.52	17.25	15.72	15.80	65.29
Dairy Calves—Male	17.11	20.14	16.04	16.15	69.44
Dairy Calves—Female	17.59	20.75	16.44	16.56	71.34
Dairy cattle—Pregnant	16.79	18.32	15.76	15.87	66.75
Dairy cattle—non-pregnant	16.25	17.93	15.26	15.57	65.01
Beef cattle—Pregnant	20.52	22.49	16.72	16.90	76.64
Beef cattle—non-pregnant	19.28	22.00	16.33	16.19	73.80
Replacement heifers	20.15	22.34	16.26	16.44	75.19
Other bovines	9.94	15.80	9.82	9.83	45.39
Breeding bulls	20.94	23.71	17.11	17.33	79.09

**Table 6 animals-13-02766-t006:** Estimated total enteric CH_4_ emission by bovine category in the seasons.

	Emission Total CH_4_ (t CH_4_)				
	Spring	Summer	Autumn	Winter	Total/Category/Year
Beef Calves	449	469	428	430	1776
Dairy Calves—Male	428	503	401	404	1736
Dairy Calves—Female	651	768	608	613	2639
Dairy cattle—Pregnant	1326	1446	1245	1253	5270
Dairy cattle—non-pregnant	252	276	237	239	1004
Beef cattle—Pregnant	698	765	569	575	2605
Beef cattle—non-pregnant	140	153	114	115	521
Replacement heifers	1007	1117	813	822	3759
Other bovines	139	221	137	138	635
Breeding bulls	105	119	86	87	395
Total/season	5194	5837	4637	4674	20,341

t, tonnes; CH_4_, methane.

## Data Availability

The data presented in this study are available on request from the corresponding author.

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
