# Peer review of "Seasonal Effect of Grass Nutritional Value on Enteric Methane Emission in Islands Pasture Systems"

_animals, 2023, doi:10.3390/ani13172766_

Round 1
Reviewer 1 Report
The study presented by Nunes et al. evaluates methane emissions in a specific region by taking into account seasonal quantity-quality variations of livestock feed in an extensive system.
The scientific content of the manuscript is quite low. However, it can be considered a useful contribution from an application point of view.
In general, the paper is reasonably well structured. The introduction defines the problem adequately, the goal of the study and the experimental design is consistent.
The materials and methods section is well structured, clear and sufficiently complete, although it could be improved.
The results are adequately reported and discussed.
Only a few details should be improved before publication:
1) Many acronyms are introduced without a previous definition and without being highlighted in bold. Review all of the text.
2) In line 105 some details about sampling should be added (how many samples per month, what spatial distribution of the plots, what surface or quantity per plot ....).
3) The part concerning the coefficients used (lines 324-344) should be inserted immediately after the presentation of the equations used (line 269).
4) All tables should be self-referencing. Insert the acronyms illustration as done for table 4.
Author Response
Dear Reviewer,
Thank you for your review, it has undoubtedly greatly improved our work and we hope that we have clarified and agreed with the changes made in accordance with your comments.
The response to each of your questions, point by point, is attached.
We remain available to clarify any further questions that may arise.

Reviewer 2 Report
The paper exhibits commendable writing style and presents valuable insights. However, there are certain areas in the experimental design, explanation, and result presentation that require further clarification and refinement.
- Chemical Analyses:
The methodology for sample collection and its subsequent pooling for chemical analysis lacks clarity. It is essential to elucidate the exact number of samples collected and whether these samples were combined for chemical analysis. While the drying process is traditionally conducted within a temperature range of 55-60°C, the paper introduces the approach of drying samples at 65°C. This deviation from the norm raises concerns about potential ramifications on chemical analysis outcomes and protein denaturation. The specific impact of a 65°C drying temperature on the accuracy of quantitative analysis remains uncertain and necessitates a more comprehensive explanation.
- Enteric Fermentation of CH4 Emission Determination:
The precise number of animals utilized for each parameter associated with CH4 emissions (as detailed in Tables 4 and 5) requires clarification. This information is pivotal in assessing the robustness of the study's findings and drawing meaningful conclusions regarding enteric fermentation.
The paper showcases a commendable writing style and contributes valuable information.
Author Response

(The authors gave the same response as above.)

Round 2
Reviewer 2 Report
Thank you to the authors for addressing my previous comments and submitting the paper's revised version. I have reviewed the manuscript again and have some further feedback, including one major comment and several grammatical and typographical issues.
My main concern revolves around the relocation of Table 3 from the previous version, now placed as Table 1. Yet, if another reviewer has already suggested this alteration, kindly overlook my comment and keep the existing arrangement intact.
As mentioned in my initial assessment, this paper needed significant English language editing and contained numerous typographical errors. It is commendable that some of these errors have been rectified in the revised version. However, I still noticed some persistent issues that continue to impact the overall readability of the paper. To assist the authors in improving the manuscript, I will detail a few of these concerns below, focusing on addressing grammatical and typographical issues more effectively.
One recurring problem is the misuse of the definite article "The" in various instances.
For example, "the pregnant dairy cows" should be revised to "pregnant dairy cows" (page 1, line 33)
"the estimation" should become "estimating" (page 1, line 23)
"each of the categories" could simply be "each category” (page 4, line 138)
changing "the one" to just "one" in some of sentences
"absolute maximum" to "the absolute maximum" (page 12, line 386)
"the cattle feed" to "cattle feed" (page 16, line 544)
On page one, line 26, it should read "were determined," and on page 4, line 96 "was utilised" should be revised to "utilized."
Also, "sub spontaneous" could be corrected to "sub-spontaneous," "medium altitude" should be "medium-altitude," and "652 million litters" should read "652 million liters." When discussing "high fat," consider using "high-fat" for consistency.
Regarding the section on statistical analyses, "were analysed" should be adjusted to "analyzed," and "following by" should be corrected to "followed by."
"fibre” could be corrected to “fiber”
for "higher altitude," use "higher-altitudes." Lastly, consider using "meters" instead of "metres."
In summary, while there have been notable improvements, I strongly recommend conducting another thorough review of the paper to address any remaining language issues and typographical errors. This additional step will ensure the manuscript 's clarity and coherence, ultimately enhancing its overall quality.
Author Response
Thanks again for the second revision of the article. Placing Table 3 as Table 1 was suggested by another reviewer. We accepted the reviewer's suggestion, although most authors did not agree with the proposed change. However, with your opinion in this review, we return to the original version, putting the table in position 3 again, as we consider that the data presented are results. Thus, the other reviewer's suggestion is not accepted, and we will put the table where it originally was.
Regarding typographical errors, we try to improve using language correctors. We also appreciate your contribution in checking errors, which were all accepted.
We hope that, after this edition and revision of the text, the manuscript will be satisfactory.

Round 3
Reviewer 2 Report
Thank you for addressing the suggested comments from the previous review. While some of the issues highlighted in the previous review have been improved, a few remaining grammatical and typographical issues still require attention.
Firstly, on page 4, line 138, it was suggested to change "each of the categories" to "each category." However, this correction has not been made, and it still reads as "each categories."
Additionally, on page 1, line 33, there is an extra space in "pregnant dairy cows" that needs to be removed for clarity.
Furthermore, the phrase "to cattle feed" contains an extra space that should be deleted to read as "to cattle feed."
The term "high fat" could be better expressed as "high-fat," but it appears that it has been changed to "highfat" without the hyphen.
The phrase "the different bovine categories" should be corrected to "different bovine categories."
Here are a few of the remaining issues:
The expression "standard error of mean" should be changed to "the mean."
Additionally, "with the value" should be corrected to "with a value."
The phrase "factors makes" should be changed to "factors make."
Lastly, "an" should be deleted from "an associated."
I highly recommend that the authors consider incorporating thorough editing into their manuscript. Such a step can significantly enhance the overall quality of the manuscript.
Author Response
Dear reviewer,
Once again, I thank you for your contribution to improving the writing of this article.
All writing has been revised with the help of an editor, I hope I have improved the article by correcting very obvious errors in English.
All your suggestions were accepted and other errors were corrected, all duly marked in red, as you can see in the attachment.
